# Do women prefer caesarean sections? A qualitative evidence synthesis of their views and experiences

**Mercedes Colomar**[1]*, **Newton Opiyo**[2], **Carol Kingdon**[3], **Qian Long**[4], **Soledad Nion**[5], **Meghan A. Bohren**[2,6], **Ana Pilar Betran**[2]

**1** Montevideo Clinical and Epidemiological Research Unit, Montevideo, Uruguay, **2** UNDP/UNFPA/UNICEF/WHO/World Bank Special Programme of Research, Development and Research Training in Human Reproduction (HRP), Department of Sexual and Reproductive Health and Research, World Health Organization, Geneva, Switzerland, **3** School of Community Health and Midwifery, University of Central Lancashire, Preston, United Kingdom, **4** Global Health Research Center, Duke Kunshan University, Kunshan, Jiangsu Province, China, **5** Faculty of Social Sciences, Sociology Department, Universidad de la República (UdelaR), Montevideo, Uruguay, **6** Gender and Women's Health Unit, Centre for Health Equity, School of Population and Global Health, University of Melbourne, Melbourne, Australia

* mcolomar@unicem-web.org

**Data Availability Statement:** All relevant data are within the manuscript and its Supporting Information files.

## Abstract

### Background

Caesarean sections (CS) continue to increase worldwide. Multiple and complex factors are contributing to the increase, including non-clinical factors related to individual women, families and their interactions with health providers. This global qualitative evidence synthesis explores women's preferences for mode of birth and factors underlying preferences for CS.

### Methods

Systematic database searches (MEDLINE, EMBASE, CINAHL, PsycINFO) were conducted in December 2016 and updated in May 2019 and February 2021. Studies conducted across all resource settings were eligible for inclusion, except those from China and Taiwan which have been reported in a companion publication. Phenomena of interest were opinions, views and perspectives of women regarding preferences for mode of birth, attributes of CS, societal and cultural beliefs about modes of birth, and right to choose mode of birth. Thematic synthesis of data was conducted. Confidence in findings was assessed using GRADE-CERQual.

### Results

We included 52 studies, from 28 countries, encompassing the views and perspectives of pregnant women, non-pregnant women, women with previous CS, postpartum women, and women's partners. Most of the studies were conducted in high-income countries and published between 2011 and 2021.

Factors underlying women preferences for CS had to do mainly with strong fear of pain and injuries to the mother and child during labour or birth *(High confidence)*, uncertainty

**Funding:** This work received funding from the UNDP-UNFPA-UNICEF-WHO-World Bank Special Programme of Research, Development and Research Training in Human Reproduction (HRP), a cosponsored programme executed by the World Health Organization (WHO).

**Competing interests:** The authors have declared that no competing interests exist.

regarding vaginal birth *(High confidence)*, and positive views or perceived advantages of CS *(High confidence)*.

Women who preferred CS expressed resoluteness about it, but there were also many women who had a clear preference for vaginal birth and those who even developed strategies to keep their birth plans in environments that were not supportive of vaginal births *(High confidence)*. The findings also identified that social, cultural and personal factors as well as attributes related to health systems impact on the reasons underlying women preferences for various modes of birth *(High confidence)*.

## Conclusions

A wide variety of factors underlie women's preferences for CS in the absence of medical indications. Major factors contributing to perceptions of CS as preferable include fear of pain, uncertainty with vaginal birth and positive views on CS. Interventions need to address these factors to reduce unnecessary CS.

## Introduction

The proportion of births by caesarean section (CS) continues to increase worldwide [1–3]. Latest trends analysis shows that between 2000 and 2015, the global average CS rate increased by 9.0% (from 12.1% to 21.1%) [3]. Although the use is not growing in all regions at the same pace and inequalities within and between countries exist, the rise is not constrained to high- and middle-income countries but also extends to low-income countries [1,3–5]. In low-income countries, the consequences of unnecessary CS use can be more severe. Substandard care and the lack of resources and skills to treat complications following CS place women and babies in these countries at a higher risk for mortality and morbidity [6].

To date, despite substantial investment in the development and testing of interventions intended to reduce unnecessary CS, few interventions have been shown to be effective [7]. An important reason underpinning the limited progress in developing effective interventions has been a failure to fully address the multifactorial determinants and complex nature of the increase, and to plan and act accordingly [8]. In particular, women and healthcare professionals both play an important role in the decision-making process for mode of birth, which occurs intertwined in complex organizations and systems with unique challenges and norms [8].

In a century with increasing medicalisation of childbirth beyond medical indications and need, multiple considerations underlie overuse of CS. Factors such as higher maternal age at birth, the increase in the prevalence of maternal obesity, in multiple birth or in the proportion of nulliparous women have been shown as major determinants of overuse. However, these factors alone cannot explain the full phenomenon. Non-medical factors such as women's fear of childbirth, social and cultural factors, clinician fear of medical litigation and sub-optimal interactions and communication between women with healthcare providers are also involved [8,9]. Understanding and addressing these other non-medical factors, their influence and dynamics among multiple stakeholders is crucial to reduce unnecessary CS [10–12].

Quantitative systematic reviews have shown that, worldwide, only a minority of women have a preference for CS, but further understanding of women's views is necessary to develop interventions that better fit women's needs and expectations [13]. In this context, we conducted a global qualitative evidence synthesis to assess women's preferences for mode of birth

and to map the factors underlying preferences for CS, including individual, health system, cultural and societal factors. Improved understanding of women's preferences and related phenomena is critical for informing the choice and design of interventions and policies to reduce unnecessary CS.

## Methods

### Search strategy and selection criteria

This review is part of a mixed-methods global review of women's and healthcare providers' preference for CS. The protocol is registered in PROSPERO (registration number CRD42016036596). Data from China and Taiwan have been published elsewhere [14] and have not been included in this paper.

Our inclusion criteria specified that studies should include women of any age, their partners, and health workers, when expressing opinions, views and perspectives regarding preferences for mode of birth, attributes of CS, societal and cultural beliefs towards mode of birth, right to choose mode of birth, and opinions on the causes of the increase in CS.

The studies had to have used qualitative methods for both data collection and analysis (e.g. thematic analysis, framework analysis, grounded theory). We included focus group interviews, individual interviews, observation, document analysis, open-ended survey questions where response data were analysed using qualitative methods, and mixed-method studies where it was possible to extract qualitative data. This criterion constituted a basic quality threshold. We excluded studies that did not meet this standard, that did not report on the methods used for data collection and analysis, or that were based on analysis of secondary data (e.g. birth registries).

We included studies conducted in any setting, such as facility-based and community-based, and across all resource settings (low, middle and high-income countries). We excluded studies published in Persian.

We searched the following databases: MEDLINE/PubMed, EMBASE, CINAHL, POPLINE, PsycINFO, Global Health Library, using a combination of the key terms 'caesarean section', 'preference', 'choice', 'knowledge', 'attitude', 'culture', 'non-medical factors', and 'health professionals-patient relations' between 1990 and 2016 without language restrictions (S1 Table). The search was updated in May 2019 for all English databases and February 2021 for MEDLINE/PubMed database. In addition, the reference lists of included studies were screened for additional studies. Two reviewers independently screened titles, abstracts, full texts and selected studies meeting inclusion criteria using Covidence. Discrepancies and uncertainties at any stage in the selection process were resolved through discussion with a third reviewer until consensus was achieved.

**Purposive sampling of included studies.** We extracted data and appraised 119 studies from 36 countries meeting the inclusion criteria. Considering the large size of the pool of eligible studies (which may limit reliable synthesis), and because qualitative evidence synthesis aims for variation in concepts rather than an exhaustive sample, we purposively sampled from the 119 studies that met our inclusion criteria (Table 1).

We created a sampling framework that took into consideration the population group (number of studies from a particular country), data richness and closeness of study data to the objectives of the review. The three-step sampling process is outlined below.

First, we selected all studies from countries which had 4 or less publications. For countries which had 5 or more studies, we mixed sampling criteria considering maximum variation sampling, data saturation and data richness. In the second step, we selected studies representing different respondents (women, family members, healthcare providers, policy makers) and

**Table 1. Number of studies mapped, sampled and included by country.**

| Countries (N = 36) | Number of mapped studies | Number of studies purposively sampled | Number of selected studies reporting women views |
|---|---|---|---|
| Argentina | 1 | 1 | 1 |
| Burkina Faso | 1 | 1 | 1 |
| Cambodia | 1 | 1 | 1 |
| Chile | 1 | 1 | - |
| Finland, Sweden and Netherlands | 1 | 1 | - |
| Germany, Ireland and Italy | 1 | 1 | 1 |
| Ghana | 1 | 1 | 1 |
| Greece | 1 | 1 | 1 |
| Japan | 1 | 1 | 1 |
| Nicaragua | 1 | 1 | - |
| Senegal | 1 | 1 | - |
| Spain | 1 | 1 | - |
| Switzerland | 1 | 1 | 1 |
| Egypt | 1 | 1 | - |
| Jordan | 1 | 1 | 1 |
| Vietnam | 1 | 1 | 1 |
| Uganda | 1 | 1 | 1 |
| France | 1 | 1 | 1 |
| Sierra Leone | 1 | 1 | 1 |
| Thailand | 2 | 2 | 2 |
| Lebanon | 2 | 2 | 2 |
| South Africa | 2 | 2 | 2 |
| Tanzania | 2 | 2 | 1 |
| Turkey | 2 | 2 | 2 |
| New Zealand | 2 | 2 | 2 |
| Canada | 5 | 5 | 4 |
| Sweden | 5 | 3 | 2 |
| UK | 11 | 7 | 7 |
| USA | 13 | 6 | 5 |
| Australia | 16 | 5 | 3 |
| Brazil | 17 | 3 | 3 |
| Iran | 21 | 6 | 4 |
| **Total** | **119** | **66** | **52** |

their different perspectives (women willing to receive a CS, women willing to have a vaginal birth, women with previous CS, providers who perceived CS as a risk and those who perceived it as a benefit). Finally, we sampled studies reporting women and family members' views and perspectives.

In parallel, we assessed data richness of the included studies, and kept only those studies with reasonable number of findings relating to the phenomena of interest; those with "thin" data or few findings relating to the phenomena of interest were excluded.

In the third step, we sampled studies until no new information was attained and data saturation was reached.

In total, we purposively sampled 66 studies fulfilling the review inclusion criteria (Table 1). 37 studies were from countries with 4 or less publications, and 29 studies were from countries with 5 or more publications. These studies represented different respondents and perspectives,

attained data saturation and had reasonable number of findings relating to the phenomena of interest.

Finally, for the findings reported in this paper, we selected 52 studies including women and family members perspectives only.

## Data extraction and management

We performed data extraction using a form specifically designed for this review. Key themes and concepts relevant to the phenomena of interest were extracted. The form was also used to extract information about: first author, date of publication, publication language, settings and demographics, study design, recruitment, data collection and analysis methods, ethics, contextual issues and conclusions. Data was extracted by one reviewer and checked by a second reviewer. Disagreements were discussed and resolved through consensus.

The extracted data focused on the key authors' interpretations of the data. Data was entered into Atlas-TI (ATLAS.ti Scientific Software Development GmbH).

## Appraisal of the methodological quality of included studies

The quality of included studies was assessed using a checklist described by Walsh and Downe [15]. The checklist assesses methodological quality and reporting by considering clarity of reporting of aims of the research, appropriateness of study design, recruitment strategy, data collection, consideration of relationship between researcher and participants, ethical issues, description of results and value of the research. Two members of the study team (MC and QL or CK) independently assessed the quality of the studies. A final statement considering each study methodological quality was made. Studies were considered to have no concerns at all, minor, moderate, major, or serious concerns depending on the level of the flaws and their impact on the credibility of study findings. Differences in the authors' appraisals were resolved through discussion. Studies were not excluded based on the results of quality assessment; quality ratings contributed to the GRADE-CERQual assessments (described below).

## Data synthesis

A conceptual coding framework was inductively developed on the basis of the data from identified themes. Data was coded by two members of the study team independently (MC and SN). The process for data coding was line by line coding to search for concepts. Studies were coded into developed concepts, and new concepts were created when deemed necessary. They then met to discuss discrepancies and determine the relevance of new codes. We conducted a thematic analysis and synthesis according to the pre-specified analysis plan outlined in the review protocol (PROSPERO registration number CRD42016036596). We tried to reflect all the dimensions of each specific theme to gain insight of the overall picture of the synthesis using a constant comparison strategy for data extraction and synthesis [16]. In brief, we followed a four-step process for data synthesis: familiarization, data extraction, coding and development of descriptive themes, and interpretive synthesis. Details of the synthesis process are presented in S2 Table.

## Assessment of confidence in the review findings

We used the GRADE-Confidence in the Evidence from Reviews of Qualitative research (GRADE-CERQual) approach to assess our confidence in the review findings [17–21]. CERQual assesses confidence in the evidence based on the following four key components.

1. Methodological limitations of included studies: the extent to which there are concerns about the design or conduct of the primary studies that contribute evidence to an individual review finding.

2. Relevance of the included studies to the review question: the extent to which the body of evidence from the primary studies supporting a review finding is applicable to the context (perspective or population, phenomenon of interest, setting) specified in the review question.

3. Coherence of the review finding: an assessment of how clear and cogent (i.e. well supported or compelling) the fit is between the data from the primary studies and a review finding that synthesizes those data.

4. Adequacy of the data contributing to a review finding: an overall determination of the degree of richness and quantity of data supporting a review finding.

Two review authors (MC and CK) independently assessed the methodological limitations of the included studies using a checklist described by Walsh and Downe [15]. The CERQual-assessments were performed by one review author (MC) and checked by at least one other review author. Ratings for all findings started as high confidence and were then downgraded if there were important concerns regarding any of the four CERQual components. The final judgement (classified as High, Moderate, Low or Very low) was based on consensus among the review authors.

This qualitative evidence synthesis is reported according to the ENTREQ Statement for Enhancing transparency in reporting the synthesis of qualitative research [22].

## Results

### Results of the search

We identified 28,386 records from electronic databases and other sources (Fig 1). Overall, 119 studies fulfilled the review inclusion criteria. Considering the large size of the pool of eligible studies and because qualitative evidence synthesis aims for variation in concepts rather than an exhaustive sample, we purposively sampled 66 studies from the 119 studies that met our inclusion criteria. 52 studies reported women and family members' perspectives and were included in this review. The selected studies were published between 2000 and 2021.

### Description of the studies

The 52 included studies were conducted in 28 different countries: nine in North America (five in USA, four in Canada); 13 in Europe (one multi-country study in Germany, Ireland and Italy; one study each in Greece, France and Switzerland; two in Sweden and seven in UK); fourteen in Asia (one each in Cambodia, Japan, Jordan and Vietnam, two in Turkey, Thailand and Lebanon, four in Iran); five in Oceania (two in New Zealand, three in Australia); seven in Africa (one each in Burkina Faso, Ghana, Sierra Leona, Uganda and Tanzania and two in South Africa); and four in Latin America (one in Argentina and three in Brazil).

Most studies were conducted between 2011 and 2021, and interviews or in-depth interviews were the most widely used data collection methods. The characteristics of the studies are presented in Table 2.

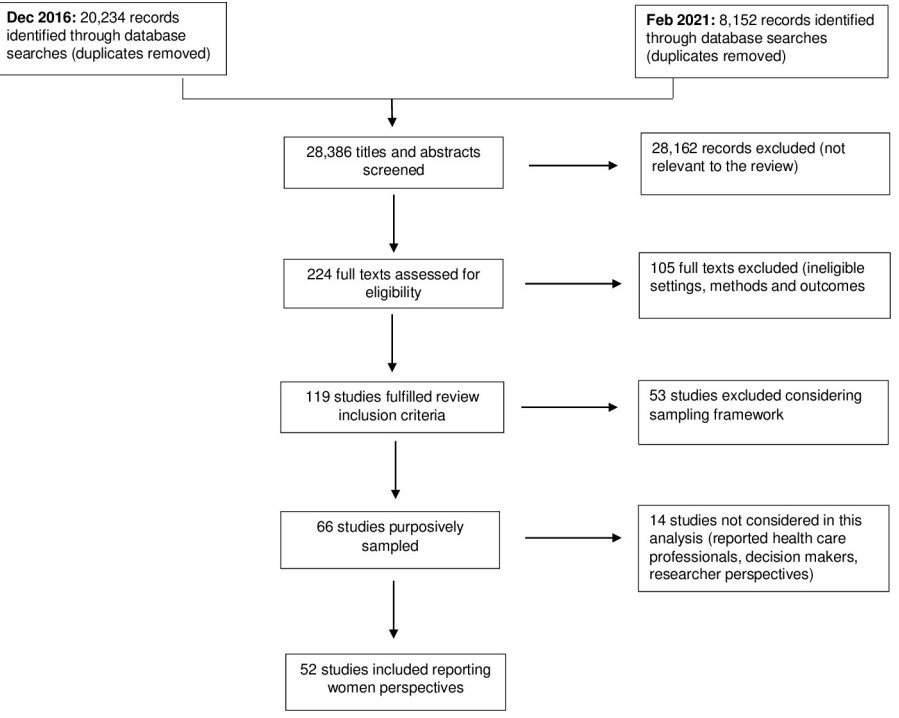

**Fig 1. PRISMA flow diagram.**

## Study settings

Four of the included studies were conducted in low-income countries (Burkina Faso, Uganda, Sierra Leona and Tanzania). Three were conducted in a lower middle-income country (Cambodia, Vietnam and Ghana). Seventeen took place in upper middle-income countries (Brazil, n = 3 studies; Iran, n = 4; Lebanon, n = 2; South Africa, n = 2; Turkey, n = 2; Argentina = 1, Jordan = 1 and Thailand, n = 2). Twenty-eight studies were conducted in high-income countries (UK, n = 7 studies; USA, n = 5; Sweden, n = 2; Australia, n = 3; Canada, n = 4; New Zealand, n = 2; one each from France, Greece, Japan and Switzerland, one was a multi-country study conducted in Germany, Ireland and Italy). These country classifications are based on the 2021 World Bank's classification of income levels [77]. Most studies (35/52) were conducted in health facilities. A summary of the study settings is presented in Table 2.

## Study participants

The studies included views of diverse groups of women: pregnant women irrespective of parity (n = 32 studies); non-pregnant women (n = 18), women with previous CS (n = 20), post-partum women (n = 11), nulliparous women (n = 11), women's partners (n = 2) and the general public (n = 2). Most of the studies included coexisting viewpoints. Details of the participants are presented in Table 2.

## Quality assessment of the included studies

We assessed most of the included studies as having minor [23] and moderate [16] methodological limitations. Six studies had serious methodological concerns. Study designs, participant recruitment and sampling strategies, methods of data collection and data analysis were appropriate in most of the studies. Most studies addressed ethical considerations and fully reported

**Table 2. Summary of characteristics of included studies.**

| Characteristic | Number of studies | Studies (References) |
|---|---|---|
| **Year of study** | | |
| 2000 to 2010 | 17 | [23–39] |
| 2011 to 2021 | 35 | [40–74] |
| **Method**[a] | | |
| Individual interview | 47 | [23–30,32–34,36–47,49–51,53–55,57–63,65–70,72–74] |
| Observation | 7 | [24,28,35,39,50,66] |
| Focus group | 11 | [30,40,47,53,54,56,58,60,67,70,71] |
| Others[b] | 6 | [28,35,36,48,64] |
| **Study region** | | |
| Africa | 7 | [44,47,53,56,63,71,73] |
| Asia | 14 | [23,27,28,33,43,50,54,55,58,61,65,69,70,74] |
| Europe | 13 | [32,34–38,42,45,52,59,60,64,66,72] |
| Latin and Central America | 4 | [24,41,51,67] |
| North America | 9 | [25,29,39,46,49,57,62,64,68] |
| Oceania | 5 | [26,30,31,40,48] |
| **Language of publication** | | |
| English | 51 | [23–40,42–74] |
| Portuguese | 1 | [41] |
| **Resource level** | | |
| Low-income | 4 | [47,53,71,73] |
| Lower middle-income | 3 | [56,61,70] |
| Upper middle-income | 17 | [23,24,27,33,41,43,44,50,51,54,55,58,63,65,67,69,74] |
| High-income | 28 | [25,26,28–30,32,34–40,42,45,46,48,49,52,56,57,59,60,62,64,66,68,72,75] |
| **Setting** | | |
| Facility based | 35 | [25,27–29,31,32,34–36,38,39,41–43,48,50,51,54–56,58–61,65,67–70,72,74,76] |
| Population based | 10 | [23,30,40,46,49,57,62,64,71,73] |
| Mixed | 6 | [24,33,37,44,52,63] |
| Unclear | 1 | [45] |
| **Participants**[c] | | |
| Pregnant women | 32 | [23,25,28,29,32,34,36–44,46,51,53–56,58–62,64,67,68,70,72,74] |
| Non-pregnant women | 18 | [23,27,29,33,49–53,57,62,63,65,66,69,71–73] |
| Women with previous CS | 20 | [26,31–35,37,39,47–51,57,60,65,69,71–73] |
| Nulliparous pregnant women | 11 | [25,27,29,36,38,42,55,59,61,67,68] |
| Postpartum women | 11 | [23,29,41,52,53,61,62,66,69,72,73] |
| Family or public members | 4 | [24,30,45,71] |

[a] Fourteen studies used more than one of the listed data collection methods [24,28,30,35,39,40,47,50,58,60,66,67,70].

[b] Includes data collection through open-ended questions in a written survey [48], field notes [28], diaries [35,36] and internet blog [64].

[c] Twenty-two studies included views from more than one of the listed participant groups [23,29,32–35,37,39,49–53,57,60–62,65,66,69,71,72].

findings. However, we judged most studies to have some concerns relating to considerations of the relationship between researchers and participants ("Researcher reflexivity"). Full details of the assessment of methodological limitation for each study is presented in S3 Table.

## Confidence in findings

The 12 review findings were graded as high confidence using the GRADE-CERQual approach. Our explanation of the GRADE-CERQual assessment for each review finding is shown in the

summary of qualitative review findings in Table 3. Full details of the evidence profile for each finding is presented in S4 Table. S5 Table presents supporting quotes for final themes, initial concepts and emergent themes.

## Summary of main findings

In this section, we report each review finding based on the CERQual assessment of confidence in each finding (Table 3). For each finding, we begin with a short overall summary and then present the overall assessment and its explanation. Full details of the evidence profile for each finding is presented in S4 Table.

Our analysis identified three types of women regarding their views on the preference and decision-making process about mode of birth: (i) women with a clearly preferred mode of birth; (ii) women who transferred the decision to the health provider; and (iii) women who wanted to discuss their options with the health provider and who were open to advice.

Among those with a clear preference, two distinct groups were identified, comprising those who preferred elective CS and those who preferred vaginal birth, and the reasons for their preferences varied. We will present the findings on the reasons underlying preferences for each mode of birth. A summary of the findings is presented in Fig 2.

Women who preferred an elective CS tended to offer an array of reasons for their decision but most expressed resoluteness about it, many noting they had always known that CS was how they would give birth. Many justified their preference on their perceived risks of vaginal birth. Findings 1 to 3 describe women beliefs underlying preferences for CS.

**Finding 1: Deep rooted fears regarding vaginal birth (High confidence).**   Some women experienced strong *fear of pain and injuries* to the mother and child during labor and birth [24,29,30,33,36,41,49,50,52,55,58–61,65,67–72,74]. It was described consistently as a reason for CS on maternal request. The findings suggested that fear of pain and of losing control over the body profoundly shapes understanding and practice in relation to increasing interventions in childbirth. In this regard, CS was reported as an easier, faster, less painful process, limiting discomfort to the mother and baby.

*Fear of uncertainty* was expressed as the fear to develop some kind of complication during labor that cannot be anticipated [25,35,36,44,48,57,59,60,65,67]. A variety of fears were mentioned, all with the potential to threaten the health of the mother and baby. Women in some studies were concerned with not knowing how long the birth process would take, how labor would be managed and how it would evolve. They expressed fear about the possibility of a long labour leading to an emergency CS, or of being hurt if giving birth to a big baby. CS was reported as a way of regaining control over the process of childbirth and as the best way of managing the uncertainty of childbirth. Studies reported women placing themselves under their physician's control as a way to feel safe.

*Fear of losing control over the body* was related to panicking and associated with tensions between norms of femininity as dainty, dignified and tidy as opposed to loss of control [44,59,61,70].

*Fear of lack of labor preparation and maternal instinct* [41,55,67,68,70,71,74]. Some women wondered about whether they would have the "maternal instinct" that will naturally guide them through childbirth or whether they would be "too posh to push". Beliefs that they would not be capable of having a normal birth were seen as the reasons that would motivate them to request an elective CS.

*Fear of negative outcomes due to vaginal birth* [24,25,29,36,50,55,58,65,70,74]. Possible complications from vaginal birth were also described as reasons for CS on maternal request. "Natural" childbirth was constructed as risky, dangerous and also unpredictable among those who

**Table 3. Summary of qualitative findings.**

| Summary of review finding | Studies contributing to review finding | Overall GRADE CERQual assessment of Confidence | Explanation of CERQual judgement |
|---|---|---|---|
| **Women beliefs underlying preferences for caesarean section as mode of birth** | | | |
| **Deep rooted fears regarding vaginal birth**<br>Women expressed having fear regarding vaginal birth which act as an underlying preference for preferring a CS as mode of birth (MOB). Fears were mainly related to labour pain, but were also interwoven with uncertainty or fear of potential negative outcomes (Concerned that the length of labor is unpredictable, failure of trial of labor and having to undergo an emergency CS, uterine rupture, adverse newborn outcomes, fear that VD will impact the tightness of vagina, fear of episiotomy, and fear of losing control over the body). In these cases fear was identified as an important factor in women's requests for CS. | [24,25,29,30,33,35,36,41,44,48–50,52,55,57–61,65,67–72,74] | High confidence | All included 27 studies across different regions (largely from high and very high developed countries (81%) [78]) and with CS rates above 25% [2] contributed to this finding. There were 8 studies that included nulliparous pregnant women, but most reported findings from women requesting CS as MOB. Overall, there were no major methodological limitations and minor concerns on coherence, relevance and adequacy. |
| **Caesarean section has advantages**<br>Women reported positive views regarding CS section which act as reasons for preferring a CS as MOB. Women referred to the idea of taking control over the birth process due to pain and anxiety. CS was also favoured because of the social advantages of scheduling birth; and perceptions of a more dignified birth experience.<br>Moreover, considering the outcomes, CS was perceived as safer for baby´s health, and better for women´s genitalia. | [23,25,29,31–36,39–46,48,50,52–55,57,59,61–65,67,70,71,74] | High confidence | All included 34 studies across different resource level regions and with CS rates ranging from 3 to 56% contributed to this finding. Four studies with serious methodological limitations and 8 with low data adequacy would reduce the confidence in the review findings. Greatest confidence was found among studies assessing women reasons for requesting a CS without medical indication, while less coherence and adequacy was found in studies that included women planning vaginal birth perspectives. |
| **Healthcare systems factors underlying preferences for caesarean section as mode of birth** | | | |
| **Quality of care**<br>Women were worried about poor quality of care if they attempt a vaginal birth.<br>Concerns regarding lack of privacy and support, as well as surrendering to HCP humiliating situation were raised. | [27,28,42–44,52,58–60,63,69,70] | High confidence | Studies supporting this finding came from countries with different levels of development and with varying CS rates. Greatest confidence was found among studies coming from countries with the higher CS rates (greater than 40%) (Turkey, Iran, South Africa, Vietnam and Greece). Overall, there were no major methodological limitations and only minor concerns on coherence, relevance and adequacy. |
| **Women beliefs underlying preferences for vaginal birth** | | | |
| **Vaginal birth is the natural way to give birth**<br>These women equated natural and demedicalized birth to be beneficial. Among the positive effects, women mentioned the benefit for newborn´s and mother´s health, quick recovery and immediate breastfeeding. | [26,32,34,35,38,39,41,43,44,46,48,50,52,53,55,57–59,62–67,71,72] | High confidence | This finding was likely to appear in highly developed countries as well as less developed ones.<br>Studies supporting this finding came mostly from countries with CS rates over 24%. 12 of these studies included women who had previously received a CS and seven included nulliparous women. Greatest confidence was found among studies including women preferring VD as MOB.<br>Overall, there were no major methodological limitations and only minor concerns on coherence, relevance and adequacy. |
| **Vaginal birth is an empowering experience**<br>The 'natural childbirth' discourse emerged as a powerful gendered technology and even enjoyable experience. Many women mentioned feeling powerful through an embodied birthing experience, when noting the strength of their bodies and what they were capable of. Empowerment also derived from the capability of confronting health professionals when standing for their right to opt for vaginal birth. | [31,32,35,39,44,46,48,50,52,57,60,62–64,67,72] | High confidence | This finding was represented by women from very high developed regions. Studies supporting this finding came mostly countries with CS rates around 30% and more. Eight of these studies (out of 15) included women who had previously received a CS. Greatest confidence was found among studies including women preferring VD as MOB.<br>Overall, there were no major methodological limitations and only minor concerns on coherence, relevance and adequacy. |

*(Continued)*

**Table 3.** (Continued)

| Summary of review finding | Studies contributing to review finding | Overall GRADE CERQual assessment of Confidence | Explanation of CERQual judgement |
|---|---|---|---|
| **Caesarean section is risky** Among women pursuing VD, CS was considered a procedure that has associated risks sufficiently large not to disregard. For that reason, these women justified it when medical indications were present or as a lifesaving procedure. | [25,31,41,45–48,53–56,58,62,64,65,67,71–73] | **High confidence** | This finding was more frequent between high and very high developed countries (87%). Most studies supporting this finding came mostly from countries with CS rates around 30% and more. Greatest confidence was found among studies including women preferring VD as MOB. Half of the studies included women with previous CS. Overall, there were no methodological limitations and only minor concerns on coherence and relevance, findings were moderately adequate. |
| **Cultural factors affecting vaginal birth** | | | |
| **The good mother imperative** There are social representations that influence women decision towards MOB. Some women mentioned the importance of the rite of passage towards motherhood when undertaking vaginal birth and of feeling birth pain that would make women more respectful. | [26,28,29,32,35,41,46,50,55,58,62–64,67,71] | **High confidence** | This finding was more frequent in studies from very high or high developed countries (93%). Studies supporting this finding came mostly from countries with CS rates over 25% and more. Greatest confidence was found among studies including women preferring VD as MOB. Overall, there were minor concerns regarding methodological limitations, coherence, relevance and adequacy. |
| **Religion advocates towards vaginal birth** Some women mentioned that their religion plays a role on their decision towards having a vaginal birth. Attempting a MOB different from vaginal birth would not be supported and even result in poor outcomes due to the contravention. | [28,43,50,53,55,58,71,73] | **High confidence** | Studies supporting this finding came mostly from countries from the Islamic world and African countries with varying rates of CS. Overall, there were minor concerns regarding methodological limitations, coherence, relevance and adequacy. |
| **CS has economic and social implications** Some women mentioned the inconvenience derived from the impact of CS on economic matters. Either because of the time it takes to recover from a CS affecting women self-sufficiency and working capacity or the impacts on family finances and the direct costs incurred by the practice itself. | [32,34,35,41,43,45,47,48,50,53,55,56,58,65,67,71–73] | **High confidence** | This finding was more frequent between high and very high developed countries (77%). Half of these studies included women who had CS already. Overall, there were no major methodological limitations and only minor concerns on coherence, relevance and adequacy. |
| **Women participation in power structures and decision making towards mode of birth** | | | |
| **Women decision towards mode of birth involves struggling to protect their right to decide** Some women expressed determination to uphold their decision regarding MOB. To defend their decision some women might design a birthing plan like home-birthing or not attending prenatal check-ups. In some cases, standing up for their decisions might imply taking responsibility for the outcome of childbirth since HCP would transfer it as a means of making women move away from chosen MOB. | [39,44,46,48,49,51,53,57,59,60,63,64,67,71] | **High confidence** | All included fourteen studies across different regions and varying development levels, with high CS rates contributed to this finding. This was utmost the case for women willing to have a VD. In seven studies women had a previous CS and were trying to avoid the next. Overall, there were no major methodological limitations and minor concerns on coherence, relevance and adequacy. |
| **Decision towards mode of birth is the result of an informed decision agreement** Some women expressed reaching an informed decision regarding the most convenient MOB for herself as a result of a healthy discussion with their HCP. | [24–26,28,29,31,32,34,35,39,42,44,46,52,59,60,63,66,67] | **High confidence** | Eighteen studies were performed in very high developed countries and one in a high developed region. Two studies were performed in South Africa including white pregnant women who could afford private care). The preferred MOB did not affect the informed consent decision. Overall, there were no major methodological limitations and minor concerns on coherence, relevance and adequacy. |
| **Mode of birth is a medical decision** Some women expressed that the MOB was finally a medical decision. Either because they lacked autonomy and were not considered by the HCP, or because they preferred giving control to others. | [24,33–35,37–39,45,48,49,51,53,58–60,67,70,73] | **High confidence** | Fourteen (out of 18) studies performed in very high developed regions contributed to this finding. In ten studies women had a previous CS. Overall, there were no major methodological limitations and minor concerns on coherence, relevance and adequacy. |

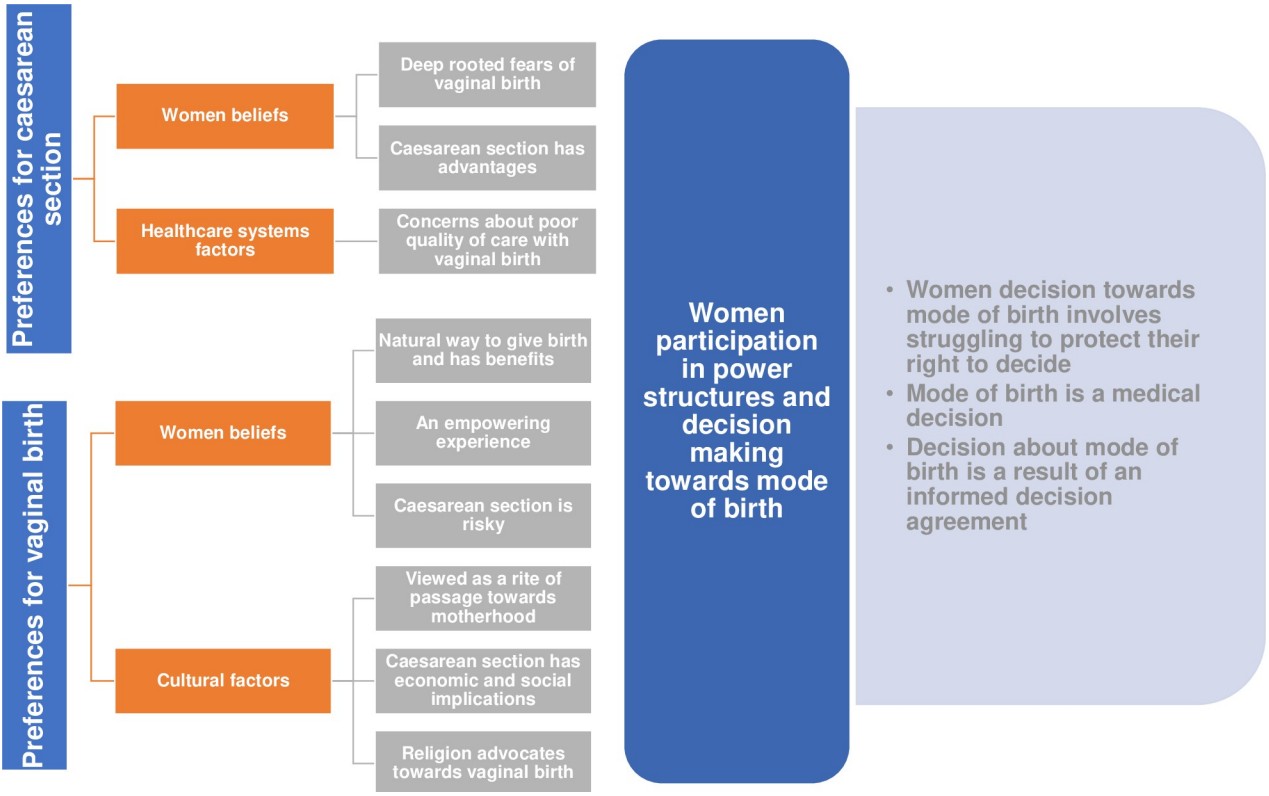

**Fig 2. Summary of findings.**

expressed preference for elective CS. Regarding perinatal outcomes, women believed that by choosing CS they were able to control the safety of their babies. Among other negative outcomes, women mentioned vaginal distortion and compromised sexual pleasure, due to the exertion of giving birth. Uterine rupture was also mentioned among women with previous CS.

The *fear that vaginal birth would impact the tightness of vagina* was a particular finding associated with the post-vaginal birth body which was characterized as 'loose' [24,50,55,58,61,67,71]. In this sense, vaginal birth was seen as antagonistic to the function of the vagina which was to provide heterosexual pleasure. For these women, availability for penetrative sex was a strong concern. In some cases, women expressed that partners' perspectives also played into the perceptions of women.

**Finding 2: Caesarean section has advantages (High confidence).**   Some women who wanted to give birth by CS had positive views on the birthing method. The reasons they stated were that *CS is a way of being under control* or that it controls pain and anxiety [23,25,31–34,36,39,41,42,44–46,48,52,54,57,59,62–65,67,70]. The attributes of CS described hinged on qualities associated with organization and control (including planning and predictability), and the avoidance of pain, the confluence of which reduced anxiety during birth. Hence, women with previous traumatic birth experience like emergency CS, previous miscarriages or having not conceived after several attempts, had high level of anxiety and felt safer with a CS.

The possibility *to plan day and time* was described hinged on qualities associated with organization and control over the timing of the birth [23,34,39,40,42,61,67,70,74].

This is related with the idea of *CS as a more civilized way to give birth* [23,25,40,42–44,50,59,62,63]. It was referred as modern and technologically advanced form of childbirth.

Hand in hand with the technology advances, there is a perception that *CS is becoming more common* [25,41,43,65]. There is a perception of a general openness to the elective procedure, based on individual rights and a growing consumer-model approach to health care that makes it a common practice.

Considering perceived positive outcomes related to CS, there is an idea that *CS heals faster* [25,43,54,70]. The incision type and speed of repair were also mentioned as long-term positive attributes of CS.

Also, women who preferred elective CS, reported to have constructed their decision in the *best interests of the baby*, largely because of what they conceived as safety associated with technologically advanced surgical birth and *less trauma for the baby* [23,29,31,33,35,36,41–45,48,53–55,57,62–65,70,74].

**Finding 3: CS would ensure better quality of care (High confidence).**   There were also health systems factors underlying women preferences for CS. Some women wanted to give birth by CS as a way to avoid *inadequate support or care during childbirth* [27,28,42–44,52,58–60,63,69,70]. They thought that a CS would be a way to avoid being humiliated and blamed by healthcare providers during vaginal birth. Moreover, some women reported not being able to be accompanied during childbirth. For this reason, a CS was a way to avoid feeling lonely and dependent on health providers who were perceived as not being supportive during childbirth.

Finally, women who preferred elective CS were influenced by different kinds of knowledge (medical and non-medical) and from multiple sources (family, friends, media, healthcare professionals), with varying degrees of influence at different time-points. Regarding non-formal sources, dramatic stories told by other women or massive audiovisual content like TV soap operas, frequently depict childbirth as agonizing. Concerning formal information sources, women who preferred elective CS reported that medical practitioners were usually in favor of CS. Given the combination of uncertainty, fear, and medical and non-medical information against vaginal birth, it appeared that many women choose elective repeat CS as a way to control some aspects of the birth process.

On the other hand, women who preferred vaginal births reproduced an essentialist and maternalistic view of femininity in which birthing babies was seen as women's primary calling and part of becoming a mother. For many women, resisting a patriarchal clinical system was a source of strength and spurred action in their decision making. Findings 4 to 9 describe the factors underlying women preferences for vaginal birth.

**Finding 4: Vaginal birth is the natural way to give birth (High confidence).**   There was a strong conviction among women who preferred vaginal births that this was the *natural way to give birth* after a normal pregnancy and that this is the way the bodies were designed [35,38,39,43,44,48,53,59,62,63,66,67,71,72]. Moreover, discomfort was considered as part of becoming a mother, and although most women desired minimal pain, they also welcomed this sensation as a unique and intrinsic part of being a mother. Some of these mothers saw a CS as an inferior form of birthing and disembodying. Some of these women already had undergone an emergency CS in previous pregnancies, and the current pregnancy was seen as the way to assert their own desire to accomplish this rite of passage.

Considering perceived positive outcomes related to vaginal birth, women preferring vaginal birth believed that by not interfering with nature, they would have a *quick recovery* thereby avoiding postsurgical complications and becoming self-sufficient faster after birth [32,34,44,48,53,57,65,67,72].

Among positive health consequences, vaginal birth was perceived to be *good for mother's health* [31,32,72,41,46,50,52,55,57,58,64]. It was considered that the process of vaginal birth provides benefits to women's body. Also, they believed that vaginal births helped them get rid of content of uterus immediately after childbirth.

*Positive health consequences were also considered for baby's health* [31,46,50,55,63,72]. Some studies described women considerations regarding the benefits of vaginal birth for babies. It was mentioned that it is better if the baby decides when to be born, since the number of drugs taken during labor and birth are reduced. Finally, some women believed that vaginal birth was a way to promote immediate contact with the baby after birth.

Hand in hand with immediate contact, vaginal birth would allow *immediate breastfeeding* [31,32,46,48,57,67]. Many studies described how women expressed the positive effects vaginal birth had on the initiation of breastfeeding favoring the bonding between mother and baby.

**Finding 5: Vaginal birth is transcendent and empowering experience (High confidence).** Vaginal birth was considered as one of the rare moments when the proscriptions of dainty and femininity can be shelved by *feelings of empowerment* through accomplishing a natural, embodied transition to motherhood. It was described as an achievement and a source of "pride" and "self-respect" [31,32,62–64,67,72,35,39,46,48,50,52,57,60].

It was reported as a unique experience, referring to the feeling of being whole or complete. Women also stated that by participating actively in the process, they could have control and avoid complications. Hence the discourse of "natural childbirth" emerged as a powerful gendered technology. Some pregnant women believed that enduring labor pain represents women's power.

In some studies the childbirth experience was described in positive terms and *considered an enjoyable experience*, qualified as relaxed [44,52,67]. In sum, women expressed to fully enjoy the childbirth experience and even childbirth pain was considered a special kind of pain defined as "unique", "beautiful", "special", "linked to life", "natural", "an expected pain", and a type of pain which is "worth suffering".

**Finding 6: For those who preferred vaginal births, CS was associated with lack of control and with fears (High confidence).** Several studies reported that women feared CS, since it was associated with maternal and neonatal complications, or with emergency intrapartum situations and medical procedures based on woman or fetus health conditions. It was also reported that being strapped to a gurney and having the abdomen cut open felt inhumane.

Many studies also reported *fear of negative outcomes due to CS* [25,45,46,53–55,62,65,67,72]. Some women reported fearing that the anesthesia might not work properly or that might be incorrectly administered. Fear of recovering from anesthesia was also reported. Also, some studies reported that women felt anesthesia could have negative effects on both mother and fetus.

*Other sequalae and problems* resulting from CS could be infection, severe abdominal adhesions, improper wound healing, long recovery time, physical side-effects (e.g. back pain and fatigue), risk of rupture of the scar from the previous CS and operation-induced adhesion [31,41,46–48,53–56,58,64,65,67,71,73].

Many studies stated that during a CS, women worried they would not have an active role since they are asleep and therefore are not able to "feel childbirth", lacking control during procedure. Choice and control were central to women's discussions around vaginal birth.

There are also cultural and social factors operating in the women preferred mode of birth, reported in findings 7–9:

**Finding 7: The good mother imperative (High confidence).** Cultural representations in various societies influenced women's tendency towards opting for vaginal birth, as a way to avoid social sanction. Some women reported facing challenges to their right to decide their preferred mode of birth based on gender-related cultural factors. The strong presence of a *selfless mothering* discourse would operate as a moral imperative of good, restricting their freedom to choose [28,31,32,41,46,55,58,62–64,67].

Some women also considered sacrifice and pain as a necessary part of the rite of passage to full motherhood. Within this framing, having a vaginal birth was associated with the notion that a woman needs to suffer to become a mother.

In many studies CS were viewed as 'coping out of your motherly duties'. Women reported that CS would be an easier alternative when physical integrity was threatened or when fear of childbirth was intense [29,35,46,50,62,63,71].

**Finding 8: Religion advocates for vaginal birth (High confidence).** Women from societies influenced by Buddhist or Islamic teachings expressed the belief that childbirth is a natural process and minimum interventions are highly valued. Many studies reported that the decision on mode of birth would be influenced by their religion, since natural birth was considered a natural phenomenon and a symbol of God's power [28,43,50,53,55,58,71,73].

**Finding 9: Caesarean section has economic and social implications (High confidence).** Many studies described the *direct and indirect costs associated with CS* procedures as potential barriers to access. Studies from Iran and Burkina Faso reported the economic implications of the direct costs of a CS which could be quite substantial for the family. Paying it reflected the love and interest of the husband to the wife and also his concern in providing comfort to her long recovery period and loss of income [43,47,71,73].

Studies from other countries reported concerns regarding indirect costs. These were mainly related to women's inability to work after a CS, which led to a *difficult economic situation* and loss of independence [43,47,53,56,71,73]. This loss of independence was a great source of stress and anxiety for many women, making them economically vulnerable and socially isolated within already challenging settings. For women who were the sole providers for their households, the economic impact of CS was especially problematic.

Finally, *inability to fulfill women family roles and responsibilities* were also identified as major disadvantages of CS [32,34,35,41,53,55,56,66,67,71,72,73]. Temporary inability to drive and the disruption to family life caused by a longer recovery period after CS were major considerations, as were the need for assistance from family members and difficulties with childcare.

Women in favor of vaginal birth also described some associated disadvantages. Most were focused on practical concerns, related to physical or medical factors, including long hours of labor, prolonged pain, exhaustion, and episiotomy stitches that were uncomfortable for some days after birth [34,45,48,50,53,58,65,72,73]. Vaginal and bladder consequences and genital complications (for sexual life) were also mentioned.

The review also identified factors relating to women's participation in power and decision-making structures towards mode of birth. These are described in Findings 10 to 12.

**Finding 10: Women's decision about mode of birth involves struggling to protect their right to decide (High confidence).** Regarding the context of the decision on how to give birth, in some studies women who preferred vaginal births described fear of going under an unnecessary CS [39,44,46,48,49,57,60,64,67,71]. Mistrust of physicians was reported and some women expressed concern about unnecessary CS performed with the purpose of rushing births in order to clear hospital beds to allow new admissions. Other studies reported healthcare providers pushed women to accept a CS considering potential risks. In these cases, the fear of blame in the event of a poor outcome, especially as this could affect the baby, highlighted the responsibility attached to decision-making [46,49,51,57,59,60]

However, many women who preferred vaginal births developed strategies to maintain their birthing plans. These women perceived birth preparation and antenatal classes as important tools in reducing the risk of CS. Some were reported to pray to cope with the situation [53,64,71], to perform regular exercise [53,60], and even not show up at antenatal check-ups so

that staff would not schedule them for CS [53]. In these situations, women actively seeks to reduce their own risk of CS.

Some women opted for homebirth as an attempt to maintain their preferred mode of birth in environments that were not supportive of vaginal birth after caesarean (VBAC) [44,49,57,59,63,64]. Some of them expressed anger about being forced into having to choose a homebirth due to lack of support for VBAC in hospitals, and into taking additional risks because of the inflexible attitude to VBAC. Many women referred to hospitals in negative terms, revealing representations suffused with fear of being subdued to protocols. However, they had favorable comments about midwives who supported them during homebirth.

**Finding 11: Decision towards mode of birth is the result of an informed decision agreement (High confidence).**   There were also studies reporting women who considered that the decision-making process about mode of birth was marked by informed consent discussions with their care providers and understanding the evidence based on research [25,28,29,31,32,34,35,39,42,44,63,66,67].

To have continuity during prenatal care with caregivers ensures good relationships and facilitates building confidence and trusting them [24,31,32,46,52,59,60,67]. However, although many women expressed a desire to be involved in the decision-making process, not all of them actively participated. It was noted that healthcare providers influence decisions about mode of birth.

**Finding 12: Mode of birth is a medical decision (High confidence).**   Some studies reported that women expressed *lack of autonomy regarding childbirth decisions* [24,38,49,58,60]. Most women had experienced little control over the decision, but accepted it because they trusted their doctor. Several women reported to have refused at first to undergo CS but were later convinced by healthcare providers. Studies also reported that, although women exercise a degree of choice, they are ultimately determined by circumstances beyond their control given their lack of knowledge and information about different modes of birth. Women reported that, regardless of how much self-education they did prior to their labor, they often still got caught up in the medical model hierarchies in ways they could not control. Thus, women's capacity to choose is severely compromised as they have little power to resist the doctor's claims to authoritative knowledge.

*Other women thought choice of mode of birth was the health worker's decision* [24,33–35,37–39,45,48,49,51,53,58–60,67,70,73]. For them, a more passive stance, including the submissive acceptance of information (disincentive to search for information, "not think", "leave it to the doctor" or "leave it to see when it comes"), can be comforting and satisfying. They value the professional who takes control of the situation because they resolved difficult personal emotions that they experienced in attempting to make an individual choice.

Finally, and independent of the preferred mode of birth or how decision was made, the most important factor in choosing a birthing method was to put the babies' needs ahead of their own. There was consensus on the premise that the end product is more important than the process. And satisfaction with the birth process was related to birth outcome "a healthy newborn" or to the lack of major complications in the early postpartum period.

## Discussion

### Summary and interpretation of findings

This qualitative evidence synthesis found that the factors underlying women preferences for CS had to do mainly with strong fear of pain and injuries to the mother and child during labour and birth *(High confidence)* and positive views on CS hinged on qualities associated with better organization and control of birth process *(High confidence)*. Women who preferred

CS expressed resoluteness about it, but there were also many women who had straightforward preferences for vaginal birth and those who even developed strategies to keep their birthing plans in environments that were not supportive *(High confidence)*. Many women who expressed concern on how providers pushed them to accept a CS considering potential risks *(High confidence)* were worried about going under an unnecessary CS *(High confidence)*, and experienced little control over the decision process *(High confidence)*. There is a need to assess to what extent the fear of pain and injuries are women related factors, or whether they are the result of providers' messages intended to perform CS with the purpose of rushing births. These findings are consistent with other studies where women felt they didn't establish balanced power relations with their healthcare providers [79–81], and where they described themselves as 'agreeing' with and 'going with the flow' of professionals' recommendations [82]. Under these circumstances it is not salient that women placed themselves under the control of their doctors as a way to feel safe, convinced that technocratic knowledge and technological advances in CS were associated with the idea of safer birth outcomes. Also, the technical language use to present information to a woman might steer the woman's decision to choose her mode of birth [82].

A companion mixed-methods systematic review from China reported similar findings regarding beliefs about CS as well as concerns about lack of support and pain-related fear with vaginal birth [14].

This review also found that social, cultural and personal factors as well as attributes related to health systems impact on the reasons underlying women preferences for various modes of birth. Women's perceptions of CS as preferable were shaped by intense fear of pain and injuries to the mother or baby during labour or birth [29,30,36,55,59,61,65], uncertainty regarding the labour process and complications, and by medical and non-medical knowledge under the technocratic model of childbirth. These findings are consistent with those of a previous review on women's request for CS [83], which found that where women request CS without medical indications, their requests are related to factors such as quality of care, fears of lack of support during birth and cultural beliefs about modes of birth.

Conversely, women who preferred vaginal births described it as an achievement and a source of "pride" in which sacrifice and pain were considered as a necessary part of the rite of passage to full motherhood [26,39,46,63]. Among these women, choice and control were central, and the discourse of "natural childbirth" emerged as a powerful gendered technology that was present mainly among women from very high developed regions [37,38]. Other cultural reasons for opting for a vaginal birth, were religious beliefs [27,39–41] and a selfless mothering discourse that would operate as a moral imperative of good, even restricting their freedom to choose [41,62,64]. Also, in more deprived contexts, direct and indirect costs associated with CS was a great source of stress and anxiety for many women leading to a preference for vaginal birth [27,45,46].

This review found that women willing to have a vaginal birth expressed concern about being subdued to an unnecessary CS performed with the purpose of rushing births. Moreover, pressures from health providers to accept a CS were also reported, and some women declared to having had to develop strategies to keep their birthing plans in order to avoid unwanted CS. These findings are consistent with those of a previous review which found that only a minority of women in a wide variety of countries and situations expressed preference for CS [46].

This review revealed that most findings are similar across low and high-income countries. Nonetheless in studies coming from high income countries some women reported that choice of mode of birth was the result of an informed decision, there were also women for whom the mode of birth was a medical decision, either because they lacked autonomy during

consultation with health providers or because they preferred giving control to the providers as a way to feel safe.

## Strengths and limitations of the review

This is the first global qualitative evidence synthesis to provide a comprehensive synthesis of women preferences for mode of birth and the motivations for the preferences, including studies across different regions worldwide (except for China which has been reported in a companion paper [14]). We excluded articles written in Persian; however, their exclusion was unlikely to bias overall findings. Our review has some limitations. It was not possible to differentiate and draw distinct themes for the different participants, as most of the studies included coexisting perspectives of different groups of women (e.g. nulliparous, women with previous CS). Also, among the included studies, some countries have larger representation and hence stronger influence on overall findings. There is also larger representation from high income countries, limiting further analysis on differences in the findings according to country income levels.

## Implications for practice and research

The findings of this review indicate that preferences for CS are mainly based on fears, uncertainty associated with vaginal birth and wrong beliefs or misconceptions regarding potential benefits of CS. Providing comprehensive health education and counseling (including psychoeducation for women with fear of childbirth) should therefore be a priority during antenatal care as recommended by WHO [84]. Also, when a request for CS arises out of maternal anxiety, health providers should explore psychosocial reasons for the requests and provide psychological based therapies (such as relaxation techniques) rather than CS [85–87]. However, in this review, we found studies reporting that, regardless of how much health education women received prior to labour, they often still got caught up in the medical model hierarchies in ways they could not control [27,36,37,41,46,51–59]. Thus, women's autonomy to choose preferred mode of birth is severely compromised. Moreover, the idea of CS as a painless mode of birth is sometimes nurtured by healthcare providers, encouraging women towards such a decision. In this context, unnecessary CS are unlikely to reduce without multifaceted strategies addressing women and health provider concerns and health system factors.

## Conclusions

A wide variety of factors underlie women's preferences for CS in the absence of medical indications. Major factors contributing to perceptions of CS as preferable include fear of pain, uncertainty with vaginal birth and positive views on CS. Health professionals should be aware of these factors and offer appropriate evidence-based interventions including prenatal birth preparation classes, psychoeducation and shared-decision making for informed birth choice. Interventions intended to optimize caesarean use should be multifaceted and address highlighted factors underlying women's preferences for CS.

## Supporting information

**S1 Table. Search strategies.**
(DOCX)

**S2 Table. Summary of the data analysis and synthesis process.**
(DOCX)

**S3 Table. Assessment of methodological limitations.**
(DOCX)

**S4 Table. Evidence profile.**
(DOCX)

**S5 Table. Supporting quotes for final themes, initial concepts and emergent themes.**
(DOCX)

## Acknowledgments

We would like to thank Tomas Allen for assistance with the literature searches.

## Author Contributions

**Conceptualization:** Carol Kingdon, Qian Long, Meghan A. Bohren, Ana Pilar Betran.

**Data curation:** Mercedes Colomar, Carol Kingdon.

**Formal analysis:** Mercedes Colomar, Soledad Nion.

**Methodology:** Meghan A. Bohren.

**Project administration:** Ana Pilar Betran.

**Supervision:** Qian Long, Meghan A. Bohren.

**Validation:** Newton Opiyo, Carol Kingdon, Qian Long.

**Writing – original draft:** Mercedes Colomar, Newton Opiyo.

**Writing – review & editing:** Carol Kingdon, Qian Long, Soledad Nion, Meghan A. Bohren, Ana Pilar Betran.

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
