## [Decision Letter · Decision Letter 0]

28 Jan 2021

PONE-D-20-32480

Do women prefer caesarean sections? A qualitative synthesis of their views and experiences

PLOS ONE

Dear Dr. Colomar,

Thank you for submitting your manuscript to PLOS ONE. After careful consideration, we feel that it has merit but does not fully meet PLOS ONE’s publication criteria as it currently stands. Therefore, we invite you to submit a revised version of the manuscript that addresses the points raised during the review process.

We look forward to receiving your revised manuscript.

Kind regards,

Eduardo Ortiz-Panozo, MD; MSc

Academic Editor

PLOS ONE

Journal Requirements:

2. Please update your search to include research published since May 2019

Additional Editor Comments:

This is a scientifically sound review of qualitative studies exploring women's preferences and beliefs on cesarean sections. I agree with reviewers concerns that some edits on methods/results sections are needed for clarity. I would suggest authors to double check consistence between methods and results, especially taking care in excluding the methods/results related to China and Taiwan data (published elsewhere). For instance, page 4 S1, are those results relevant to the present review?

Reviewers' comments:

Reviewer's Responses to Questions

**Comments to the Author**

1. Is the manuscript technically sound, and do the data support the conclusions?

Reviewer #1: Yes

Reviewer #2: Yes

2. Has the statistical analysis been performed appropriately and rigorously? 

Reviewer #1: N/A

Reviewer #2: N/A

3. Have the authors made all data underlying the findings in their manuscript fully available?

Reviewer #1: Yes

Reviewer #2: Yes

4. Is the manuscript presented in an intelligible fashion and written in standard English?

Reviewer #1: Yes

Reviewer #2: Yes

5. Review Comments to the Author

Reviewer #1: In general, the text is a very important contribution to the literature on reproductive health. Here are some comments on the text in order to make it easier and more understandable to read. In the same way, I make some comments about methodological doubts that arise when reading the document.

The methods section is well addressed and fulfil all major questions, also the flow diagram is useful for easy review.

Respect to the “appraisal of the methodological quality of included studies” section:

How many studies wherein which category, especially those that had more problems categorized as “serious concerns”.

“Results of the search” may be included on the “Search strategy and selection criteria” section.

“Data synthesis” section would be more useful on the results section following the “confidence in the findings” section.

On P5 it reads “We included studies that were published in English, Spanish and Portuguese, based on the languages spoken by the review authors.” but later in the document on P31 authors say “We considered both English and non-English published literature and assessed abstracts and full-text articles written in English and Portuguese” It is not clear if studies in Spanish were or not included. Could be useful to clarify how many studies in which language was analyzed.

“Conclusions sections” - It is a good closing paragraph, but it seems as an uncomplete section, may be better to link it to the previous recommendation section,

References 9, 6, 11, 13, and 14 are literature reviews included in the synthesis, how do the data limitations where worked? For example, in most literature reviews fewer qualitative data for analysis is included, how does this affect the data, why did you decide to include this, how the information inside this reviews was classified. This is the major concern I have related to the text.

Reviewer #2: • Overall: An important topic with contribution to the current worldwide concerns around the rising rates of caesarean sections.

• Introduction: Very good introduction highlighting the need to identify factors influencing high rates of CSs. There are descriptions around clinical and non-clinical factors. However, there needs to be an emphasis on the rationale for this systematic review focusing on ‘if and how women’s views and their preferences for CS play a role’.

• Methods: Very well explained methods and selection of studies. A few minor edits/typos

o Page 9 Second line ‘selected 46 studies including women and family members perspectives only’ should be ‘selected 46 studies including women’s and family members’ perspectives only’.

o Page 9 second last sentence under data extraction and management. ‘authors’ should be ‘authors’’

• Results: A few comments

o A total of 46 studies are included for analysis. However, there are only 44 studies in table 2 for study region.

o Page 15: Study participants – ‘women’s partners (n=2) and the general public (n=2)’ is different to the information provided in Table 2.

o A diagrammatic presentation of the key findings from the systematic review will be valuable on this topic to see all the emerging issues in one figure.

o Were there any differences in findings among High-income and low-income countries?

• Discussion: The readers of this paper would benefit from a discussion of the following

o Critical analysis on the influence of ‘clinicians’ perspectives’ on women’s preferences and requests to birth by CS, and women ‘going with the flow’ of professionals’ recommendations. A part of it is presented in finding 10, 11, 12. However, this needs further discussion on

how clinicians’ views and beliefs influence women’s preferences and requests

how clinicians present the information to women

and the impact of it on women’s choice of mode of birth

o A recently published paper has presented a critique on women’s requests and preferences and their contribution to rising rates of CS.

https://doi.org/10.1016/j.midw.2020.102765

• Conclusion: The conclusion section is too short and needs to be developed further.

6. PLOS authors have the option to publish the peer review history of their article (what does this mean?). If published, this will include your full peer review and any attached files.

Reviewer #1: No

Reviewer #2: No

---

## [Author Response · Author response to Decision Letter 0]

1 Mar 2021

Editor Comments regarding Journal Requirements:

Response: Thank you for noting it. We have now updated and ensured all sections of the manuscript comply with PLOS ONE style requirements. 

2. Please update your search to include research published since May 2019

Response: We updated PubMed searches and identified 10 new studies. We reviewed the findings from the new studies against the main findings summarised in Table 3. Overall, the findings from the new studies do not alter the main findings and are listed in Figure 2. 

3. This is a scientifically sound review of qualitative studies exploring women's preferences and beliefs on cesarean sections. I agree with reviewers concerns that some edits on methods/results sections are needed for clarity. I would suggest authors to double check consistence between methods and results, especially taking care in excluding the methods/results related to China and Taiwan data (published elsewhere). For instance, page 4 S1, are those results relevant to the present review?

Response: Thank you for pointing this out. We have now edited the Methods and Results sections as appropriate (e.g. made clear that we excluded studies published in Persian, see Page 5) and responded to the reviewer suggestions about reporting aspects of data synthesis and search results (see our Response to Reviewer 1, Comment no. 2 and 3). Also, we have now reviewed the whole manuscript and ensured consistency in reporting across the different sections. We also excluded the results obtained from the search in the Chinese database from figure 1.

Reviewer 1 

1. Respect to the “appraisal of the methodological quality of included studies” section:

How many studies wherein which category, especially those that had more problems categorized as “serious concerns”.

Response: The “appraisal of the methodological quality of included studies” section aims to explain the methods implemented for the quality appraisal. Findings of the quality assessment are reported in the Results section, under “Quality assessment of the included studies” (Page 15-16). The number of studies judged to have serious methodological concerns now specified (Page 15, Line 276). Further details including number of studies with limitations classified as “minor”, “moderate” or “serious” can be found in S3 Table. 

2. “Results of the search” may be included on the “Search strategy and selection criteria” section.

Response: The “Results of the search” was left under the “Results” section considering PRISMA checklist reporting guidance (see PRISMA checklist in http://www.prisma-statement.org/)

3. “Data synthesis” section would be more useful on the results section following the “confidence in the findings” section.

Response: The “data synthesis” section explains the methods implemented to accomplish the third step in the thematic synthesis process, as presented in S2 Table (Summary of data analysis and synthesis process). Our reporting is in line with PRISMA guidance.

4. On P5 it reads “We included studies that were published in English, Spanish and Portuguese, based on the languages spoken by the review authors.” but later in the document on P31 authors say “We considered both English and non-English published literature and assessed abstracts and full-text articles written in English and Portuguese” It is not clear if studies in Spanish were or not included. Could be useful to clarify how many studies in which language was analyzed.

Response: We agree it was not clear enough. We meant to state that the initial search did not exclude papers written in Spanish or Portuguese, but finally since we did not retrieve any paper written in Spanish, we didn’t assess any. We have now rephrased the sentence in the “methods” section to “We excluded studies published in Persian”. We also edited the “Strengths and limitations of the review” section (line number 631) to ensure consistency in reporting. Finally, in table 2 we have now included information regarding each paper language. 

5. “Conclusions sections” - It is a good closing paragraph, but it seems as an uncomplete section, may be better to link it to the previous recommendation section,

Response: We have broadened our conclusion (below text in italics added).

“A wide variety of factors underlie women’s preferences for CS in the absence of medical indications. Major factors contributing to perceptions of CS as preferable include fear of pain, uncertainty with vaginal birth and positive views on CS. Health professionals should be aware of these factors and offer appropriate interventions including prenatal birth preparation classes, psychoeducation and shared-decision making for informed birth choice, appropriate communication and respectful care. Interventions intended to reduce unnecessary caesarean section should be multifaceted and address highlighted factors underlying women’s preferences for CS.” 

6. References 9, 6, 11, 13, and 14 are literature reviews included in the synthesis, how do the data limitations where worked? For example, in most literature reviews fewer qualitative data for analysis is included, how does this affect the data, why did you decide to include this, how the information inside this review was classified. This is the major concern I have related to the text.

Response: Thank you for noticing this inconsistency. This is a mistake on our part. The references of the reviews are not actually included in the synthesis. Appears you got the references from the previously submitted table 2 which was not correct. That table was prepared in a separate document, and the references did not automatically update when we copied it into the main manuscript. We apologize for the error and the confusion caused. All the references from table 2 have now been updated and are consistent with the methods and inclusion/exclusion criteria representing the included studies. Similarly, table 3 presenting the summary of qualitative findings does not contain references of reviews.

Reviewer #2: 

Overall: An important topic with contribution to the current worldwide concerns around the rising rates of caesarean sections.

1. Introduction: Very good introduction highlighting the need to identify factors influencing high rates of CSs. There are descriptions around clinical and non-clinical factors. However, there needs to be an emphasis on the rationale for this systematic review focusing on ‘if and how women’s views and their preferences for CS play a role’.

Response: Thank you for your suggestion. We feel the rationale and justification for the qualitative synthesis conducted (stated below) is adequate and aligned to the synthesis methods implemented.

“Quantitative systematic reviews have shown that, worldwide, only a minority of women have a preference for CS, but further understanding of women’s views is necessary to develop interventions that better fit women’s needs and expectations (13). In this context, we conducted a global qualitative evidence synthesis to assess women’s preferences for mode of birth and to map the factors underlying preferences for CS, including individual, health system, cultural and societal factors. Improved understanding of the phenomena is critical for informing the choice and design of interventions and policies to reduce unnecessary CS.”

2. Methods: Very well explained methods and selection of studies. A few minor edits/typos

o Page 9 Second line ‘selected 46 studies including women and family members perspectives only’ should be ‘selected 46 studies including women’s and family members’ perspectives only’.

Response: Thank you for noting it. It was edited as suggested. 

3. Page 9 second last sentence under data extraction and management. ‘authors’ should be ‘authors’’

Response: Thank you for noting it. It was edited as suggested. 

Results: A few comments

4. A total of 46 studies are included for analysis. However, there are only 44 studies in table 2 for study region.

Response: Thank you for noting it. It was verified and the reporting is now consistent. Consistency between study region information provided in table 2 was also verified with information provided in Table 1 (Number of studies mapped, sampled and included by country).

6. Page 15: Study participants – ‘women’s partners (n=2) and the general public (n=2)’ is different to the information provided in Table 2.

Response: Thank you for noting it. It was verified and the information provided is now correct. It is actually one study including women partners’ views.

7. A diagrammatic presentation of the key findings from the systematic review will be valuable on this topic to see all the emerging issues in one figure.

Response: We have included a Figure (Fig 2) presenting the key findings as suggested. (page 30 line 568)

8. Were there any differences in findings among High-income and low-income countries?

Response: Where present, differences in findings across low and high-income countries are highlighted in Table 3 (under the “Explanation of CERQual judgement”). We have also amended the text in the discussion section referring to specific finding linked to high income countries (line 620). The new text reads as follows: 

“This review revealed that most findings are similar across low and high-income countries. Nonetheless, in high income countries some women reported that choice of mode of birth was the result of an informed decision, and there were women for whom the mode of birth was a medical decision, either because they lacked autonomy during consultation with health providers or because they preferred giving control to the providers as a way to feel safe”. 

Considering your point, we also included in this revision an additional limitation to the “Strengths and limitations of the review” subsection, stating that: “There is also larger representation from high income countries, limitation further analysis on differences on the findings according to country income levels” 

9. Discussion: The readers of this paper would benefit from a discussion of the following

Critical analysis on the influence of ‘clinicians’ perspectives’ on women’s preferences and requests to birth by CS, and women ‘going with the flow’ of professionals’ recommendations. A part of it is presented in finding 10, 11, 12. 

However, this needs further discussion on: 

how clinicians’ views and beliefs influence women’s preferences and requests

how clinicians present the information to women

and the impact of it on women’s choice of mode of birth

Response: We understand the view of the reviewer as clinicians are crucial stakeholders in the decision-making for mode of birth. We haven’t discussed how clinicians’ views and beliefs influence women’s preferences and requests, how they present the information and impact on women’s choices of mode of birth, because this paper does not consider health professionals perspectives. The views and beliefs of the healthcare professionals are and independent systematic review and manuscript Nevertheless, we understand that the interaction between women and clinicians’ views are complex and important in the decisional process and we have included the below paragraph considering your suggestion. 

“Many women expressed concern on how providers pushed them to accept a CS considering potential risks (High confidence), were also worried about undergoing unnecessary CS (High confidence), and experienced little control over the decision process (High confidence). There is need to assess to what extent the fear of pain and injuries are women related factors, or whether they are the result of providers’ messages looking forward to perform CS with the purpose of rushing births or avoiding complications that can results in litigation. These findings are consistent with other studies where women felt they didn’t establish balanced power relations with their healthcare providers (74–76). Under these circumstances, it is not salient that women placed themselves under the control of their doctors as a way to feel safe, convinced that technocratic knowledge and technological advances in CS were associated with the idea of safer birth outcomes”

10. A recently published paper has presented a critique on women’s requests and preferences and their contribution to rising rates of CS. 

https://doi.org/10.1016/j.midw.2020.102765

Response: We have noted and thank the reviewer for sharing the related synthesis of reviews comparing influence of women’s request and preference on caesarean section. 

11. Conclusion: The conclusion section is too short and needs to be developed further.

Response: We have broadened our conclusion (below text in italics added).

“A wide variety of factors underlie women’s preferences for CS in the absence of medical indications. Major factors contributing to perceptions of CS as preferable include fear of pain, uncertainty with vaginal birth and positive views on CS. Health professionals should be aware of these factors and offer appropriate interventions including prenatal birth preparation classes, psychoeducation and shared-decision making for informed birth choice, appropriate communication and respectful care. Interventions intended to reduce unnecessary caesarean section should be multifaceted and address highlighted factors underlying women’s preferences for CS.”

---

## [Decision Letter · Decision Letter 1]

12 Apr 2021

PONE-D-20-32480R1

Do women prefer caesarean sections? A qualitative evidence synthesis of their views and experiences.

PLOS ONE

Dear Dr. Colomar,

Thank you for submitting your manuscript to PLOS ONE. All the comments from last round has been addressed. However as requested in your last email (March 4th) please send a new version with the amendment to the FUNDING section of the manuscript so I can proceed to accept your work for publication.

We look forward to receiving your revised manuscript.

Kind regards,

Eduardo Ortiz-Panozo, MD; MSc

Academic Editor

PLOS ONE

Journal Requirements:

Reviewers' comments:

Reviewer's Responses to Questions

**Comments to the Author**

1. If the authors have adequately addressed your comments raised in a previous round of review and you feel that this manuscript is now acceptable for publication, you may indicate that here to bypass the “Comments to the Author” section, enter your conflict of interest statement in the “Confidential to Editor” section, and submit your "Accept" recommendation.

Reviewer #1: All comments have been addressed

Reviewer #2: All comments have been addressed

2. Is the manuscript technically sound, and do the data support the conclusions?

Reviewer #1: Yes

Reviewer #2: Yes

3. Has the statistical analysis been performed appropriately and rigorously? 

Reviewer #1: N/A

Reviewer #2: N/A

4. Have the authors made all data underlying the findings in their manuscript fully available?

Reviewer #1: Yes

Reviewer #2: Yes

5. Is the manuscript presented in an intelligible fashion and written in standard English?

Reviewer #1: Yes

Reviewer #2: Yes

6. Review Comments to the Author

Reviewer #1: All comments from has been addressed, therefore I recommend for publication. This text will contribute to the reproductive health literature and it addresses a very important issue for different countries in which C-sections had become the norm.

Reviewer #2: (No Response)

7. PLOS authors have the option to publish the peer review history of their article (what does this mean?). If published, this will include your full peer review and any attached files.

Reviewer #1: No

Reviewer #2: No

---

## [Author Response · Author response to Decision Letter 1]

14 Apr 2021

Thank you for inviting us to submit a revised version of the manuscript. The funding section was now updated and we ensured that figure 2 was referred in the text.

---

## [Editor Report · Decision Letter 2]

20 Apr 2021

Do women prefer caesarean sections? A qualitative evidence synthesis of their views and experiences.

PONE-D-20-32480R2

Dear Dr. Colomar,

We’re pleased to inform you that your manuscript has been judged scientifically suitable for publication and will be formally accepted for publication once it meets all outstanding technical requirements.

Kind regards,

Eduardo Ortiz-Panozo, MD; MSc

Academic Editor

PLOS ONE
---

## [Editor Report · Acceptance letter]

23 Apr 2021

PONE-D-20-32480R2 

Do women prefer caesarean sections? A qualitative evidence synthesis of their views and experiences. 

Dear Dr. Colomar:

I'm pleased to inform you that your manuscript has been deemed suitable for publication in PLOS ONE. Congratulations! Your manuscript is now with our production department. 

Kind regards, 

on behalf of

Dr. Eduardo Ortiz-Panozo 

Academic Editor

PLOS ONE